# Biosensors Based on Advanced Sulfur-Containing Nanomaterials

**DOI:** 10.3390/s20123488

**Published:** 2020-06-19

**Authors:** Chunmei Li, Yihan Wang, Hui Jiang, Xuemei Wang

**Affiliations:** State Key Laboratory of Bioelectronics, National Demonstration Center for Experimental Biomedical Engineering Education, School of Biological Science and Medical Engineering, Southeast University, Nanjing 210096, China; li_chunmei@foxmail.com (C.L.); yihanwangxynu@163.com (Y.W.); sungi@seu.edu.cn (H.J.)

**Keywords:** sulfur-containing nanomaterials, metallic sulfide nanomaterials, sulfur-containing quantum dots, biosensors

## Abstract

In recent years, sulfur-containing nanomaterials and their derivatives/composites have attracted much attention because of their important role in the field of biosensor, biolabeling, drug delivery and diagnostic imaging technology, which inspires us to compile this review. To focus on the relationships between advanced biomaterials and biosensors, this review describes the applications of various types of sulfur-containing nanomaterials in biosensors. We bring two types of sulfur-containing nanomaterials including metallic sulfide nanomaterials and sulfur-containing quantum dots, to discuss and summarize the possibility and application as biosensors based on the sulfur-containing nanomaterials. Finally, future perspective and challenges of biosensors based on sulfur-containing nanomaterials are briefly rendered.

## 1. Introduction

As a by-product of oil refining and natural-gas purification, sulfur usually exists in the form of sulfide, sulfate or elementary substance in nature and is one of most abundant elements [1,2,3]. Since the discovery of sulfur, research involving sulfur has always been at the center of scientific research topics. Researchers have dedicated to exploiting the wide applications of sulfur. Until now, sulfur has been important in our daily life with a wide variety of applications, such as vulcanization of rubber, being cathode of rechargeable battery, raw material for fertilizer, insecticide, plastic and gunpowder [1,4,5,6]. Under the right conditions, sulfur is also well-known to form compounds with numerous other elements (e.g., lead, calcium or iron), and even form sulfur-containing nanomaterials.

A variety of sulfur-containing nanomaterials have been reported, such as metallic sulfide nanomaterials, sulfur-containing quantum dots, sulfur-containing organosilicon compounds, and lithium sulfide materials [7,8,9,10]. Sulfur-containing nanomaterials (e.g., metallic sulfide nanomaterials and sulfur-containing quantum dots) exhibit excellent properties, such as nanometric scale, water-dispersible, non-toxicity, excellent catalytic activity, conductivity, photoactivity and fascinating optical properties, and they have proven useful in many biomedical applications including imaging and sensing [7,8]. As known, metallic sulfide nanomaterials have been used as photoactive materials which can generate photocurrent excited by light in biosensing systems. Some sulfur-containing quantum dots can stably bind with biomolecules or other nanomaterials due to their functional groups (e.g., amino, carboxyl and sulfhydryl groups) as common reaction sites within biological systems. This allows their versatile roles as functional biomaterials in biosensor, biolabeling, drug delivery and diagnostic imaging technology [7,11,12,13]. Moreover, some sulfur-containing quantum dots (e.g., Ag_2_S quantum dots), exhibit high absorption in near-infrared (NIR) region, which enables their applications in bioimaging, biolabeling, deep tissue imaging, diagnostics and photodynamic therapy [7]. In this review, we will summarize the most recent advances on the applications of biosensors fabricated based on sulfur-containing nanomaterials and their composites (Scheme 1). Since there are too many sulfur-containing nanomaterials, it is impossible to provide a comprehensive overview of all sulfur-containing nanomaterials in a mini-review. Thus, we aim to provide two categories of sulfur-containing nanomaterials, i.e., metallic sulfide nanomaterials and sulfur-containing quantum dots. Concretely speaking, the metallic sulfide nanomaterials include binary, ternary, quaternary and non-metallic/metallic hetero-sulfides. The sulfur-containing quantum dots consist of sulfur, sulfide and sulfur-doped quantum dots. Firstly, we will briefly introduce various kinds of metallic sulfide nanomaterials or sulfur-containing quantum dots and summarize their synthetic approaches, respectively. Then, we will discuss the possibility as biosensors of the two categories, respectively. We also summarize the applications of biosensors based on metallic sulfide nanomaterials or sulfur-containing quantum dots, respectively. Lastly, the future perspectives and challenges of biosensors based on metallic sulfide nanomaterials or sulfur-containing quantum dots are briefly rendered.

## 2. Metallic Sulfide Nanomaterials

### 2.1. Generalities

Metal sulfides contain chemical bonding of one or more sulfur atoms (S) to a metal (M) [7]. They can be broken down into four main categories: binary, ternary, quaternary, and non-metallic/metallic hetero-sulfides, which can be denoted by the chemical formulas of M_x_S_n_, M_x_M’_y_S_n_, M_x_M’_y_M”_z_S_n_ and M_x_A_i_B_j_…C_k_S_n_ (A, B, C = non-metallic atoms), respectively. It should be noted that actually metal sulfide nanomaterials also include metal sulfide quantum dots, which will be illustrated in the section of “sulfur-containing quantum dots” below.

Binary sulfides. Binary sulfides (M_x_S_n_, e.g., MoS_2_, NiS, Cu_2_S, Bi_2_S_3_, CuS, SnS, In_2_S_3_ and Ag_2_S) [14,15,16,17,18], containing one type of metal and S atom in their chemical formulas, have received substantial attention for their applications in fields of sensing [19,20], photothermal therapy [21], antibacterial and antifungal activity [22], ablation therapy [23], optoelectronics [24], photovoltaic [25,26] and magnetic device [27].

Among binary sulfides, transitional metal disulfides, such as ZnS, CuS, CdS, MoS_2_, WS_2_ and NiS [19,28,29,30,31,32], have been widely studied during the past few years as new members of 2-dimensional (2-D) family. The transitional metal disulfides are typical layered materials with sandwich-like structures, where metal atoms sandwich between two layers of S atoms by strong chemical bonds and two layers of S atoms are stacked together by weak van der Waals forces [25]. Similar to graphene, graphene oxide and other 2-D materials, transitional metal disulfides are promising biosensing materials due to their excellent properties, such as large active surface areas, and the suitable bandgaps. Large active surface areas in their sandwich-like structure can provide abundant active sites to establish particular bonds between layers and biological analytes, then target specific biomolecules, and finally promote specific reactions on the surface of 2-D transitional metal disulfides. In addition, suitable bandgaps endow transitional metal disulfides with advantageous optoelectronic properties, which can improve sensitivity in electrochemical, electrochemiluminescence (ECL) and photoluminescence (PL) biosensors.

Ternary sulfides. Ternary sulfides (M_x_M’_y_S_n_, e.g., Ni_3_In_2_S_2_, Ni_3_Tl_2_S_2_ and NiCo_2_S_4_ [33,34]) contain two types of metals and S atoms in their chemical formulas. By changing the two types of metals, tuning the atomic ratios of metal or S atoms, researchers have validated different properties of ternary metal sulfides [35,36]. These ternary sulfides exhibit more flexible properties arising from their enhanced chemical and structural freedoms. These increased freedoms endow ternary sulfides with more suitable chemical and physical properties to satisfy a certain requirement, such as for more sophisticated biosensors. The bandgaps of some ternary sulfides vary those of binary sulfides [37,38], and the changed bandgaps make ternary sulfides more suitable for application in biosensors.

Quaternary sulfides. Quaternary sulfides contain three types of metal and S atoms in their chemical formulas, which have common composition of M_x_M’_y_M”_z_S_n_ where M, M’, M” = Zn, Cd, Mn, Hg, Cu, Ge, Sn, Cd, Fe, Co or Ba [39,40,41,42,43,44].

Non-metallic/metallic hetero-sulfides. Non-metallic/metallic hetero-sulfides have attracted considerable interests recently, which contain not only metal and S atoms but also other non-metallic atoms in their chemical formulas. For example, phosphor-chalcogenides [45,46] (e.g., Pd_3_(PS_4_)_2_) are an emerging class of non-metallic/metallic hetero-sulfides.

Literature [7,23,25,34,37,47,48,49,50,51,52] has reported that different morphologies of metallic sulfide nanomaterials such as nanowires, nanoplates, hollow ellipsoid, nanotubes, hollow spheres, nanorods, flowerlike structures, core-shell nanoparticles, nanoribbons and complex hierarchal micro/nanostructures been synthesized. Different synthetic approaches, such as dip-coating, chemical vapor deposition (CVD), aqueous one-step wet chemistry, hydrothermal, coprecipitation, exfoliation, sputtering, solid-state reaction, ball-milling and biosynthetic methods, were used to synthesize metal sulfide nanomaterials (shown in Table 1). Even same metal sulfide nanomaterials synthesized with different methods may exhibit different properties and be applied in different areas [30]. Therefore, it is necessary to find the suitable synthetic techniques for metallic sulfide nanomaterials. For convenience, we can also buy metallic sulfides in the market for experimental research. After investigation, most of binary sulfides (e.g., WS_2_ powders, Cu_2_S powders, ZnS powders and CuS powders) have been available in the market, but ternary, quaternary, and non-metallic/metallic hetero-sulfides have not been available in the market.

### 2.2. Applications in Biosensors

Metallic sulfide nanomaterials, as important and emerging materials, have arisen quickly in the area of biosensing due to their specific properties, namely, nanometric scale, water-dispersibility, large specific surface area, excellent catalytic activity, conductivity, biosafety, PL quenching abilities, photoactivity, and fascinating optical properties [7,48,61].

Catalysis: The metallic sulfide nanomaterials had excellent catalytic activity due to their high density of active sites, which can be used as modifiers in fabrication of novel biosensors [39,48]. Catalytic activities by the unsaturated sulfur commonly localize on the edge sites of metallic sulfide nanomaterials, which leads to fast heterogeneous electron transfer rate at the edge sites and enhanced catalytic activities [62].

Conductivity: The metallic sulfide nanomaterials had high electronic conductivity due to their low bandgaps, which make them be used as electrode materials in fabricating biosensors via electrochemical or ECL assays. For example, Chen et al. [20] have constructed a non-enzymatic glucose biosensor based on the high electronic conductivity of NiS nanospheres.

Biosafety: Some metallic sulfide nanomaterials, such as silver, copper and iron sulfides, were non-toxic [7]. These non-toxic metallic sulfide nanomaterials showed good biocompatibility in vitro, thus they could be used to fabricate biosensors.

PL quenching effect: Quencher is one of important component in PL (especially, fluorescence) sensing platforms for detection of biomolecules. In our previous work, we have demonstrated graphene possesses unprecedented PL quenching abilities [63]. Just like graphene, some representative 2-D metallic sulfide nanomaterials with 2-D layer structure also exhibited PL quenching abilities. These metallic sulfide nanomaterials with PL quenching abilities were suitable for constructing PL biosensors via PL quenching effect. Wang et al. [64] have used CuS nanoplates as quencher for fast, sensitive and selective detection of DNA via fluorescence quenching effect.

Photoactivity: some metallic sulfide nanomaterials were photoactive materials which can convert light illumination into electrical signals. When excited by light, electrons of metallic sulfide nanomaterials transferred from valence band to conduction band, resulting in the separation of photogenerated electrons and holes [65,66].

Fascinating optical properties: Some metallic sulfide nanomaterials, especially 0-D metal sulfide nanomaterials (namely, sulfide quantum dots), emitted high fluorescence. In comparison with organic fluorophores or alloy nanoclusters, metallic sulfide nanomaterials are superior as biomarkers due to their water-dispersibility, long lifetimes, resistance to photobleaching and biosafety [67,68]. Moreover, some metallic sulfide nanomaterials (e.g., Ag_2_S quantum dots), emitted tunable fluorescence in near-infrared (NIR) region, which enabled their applications in bioimaging, biolabeling, deep tissue imaging, diagnostics and photodynamic therapy [69]. Even some metallic sulfide nanomaterials (e.g., EuS nanocrystals) have been used as ECL luminescent signal source, which endowed them with possibility of fabricating ECL biosensors [70].

Based on these specific properties of metallic sulfides that can be used to prepare biosensors, much research efforts have been devoted to developing biomolecule sensors for understanding physiological or pathological functions of biomolecules in living body or cells. In recent years, metal sulfide nanomaterials mainly have been applied to establish four types of biosensors, including electrochemical, photoelectrochemical (PEC), ECL and PL biosensors, for probing various types of biomolecules.

#### 2.2.1. Electrochemical Biosensors

Metallic sulfide nanomaterials have been applied to establish electrochemical biosensors commonly due to their properties of conductivity, catalysis and biosafety. For example, Guo et al. [34] have developed a nonenzymatic glucose biosensor utilizing hierarchically porous NiCo_2_S_4_ nanowires due to their novel catalytic properties (Figure 1A). In synthetic processes, using electrospum graphitic nanofiber (EGF) as skeletons, NiCo_2_S_4_ nanowires were grown on the EGF toward different directions to decrease the agglomeration of NiCo_2_S_4_. In addition, the NiCo_2_S_4_ nanowires on EGF were core-shell structures with rough surface and polycrystalline nature. When applied to glucose determination, the skeletons of EGF and core-shell structures of NiCo_2_S_4_ enlarged effective surface to interact with glucose in solution and suppled more electrochemical active sites for accelerating glucose oxidation. Due to good sensing performances and biocompatibility toward glucose, an electrochemical biosensor based on NiCo_2_S_4_/EGF system was proposed with fast response (reaching a stable state within 5 s), a wide linear range (0.0005~3.571 mM, R^2^ = 0.995) and low detection limit (0.167 μM, S/N = 3) via amperometric strategy.

Wang et al. [62] have also constructed a high-performance electrochemical platform for biosensing glucose and lactate in sweat based on their catalytic properties and conductivity of MoS_2_ nanocrystals. The MoS_2_ nanocrystals displayed enhanced catalytic activities and fast heterogeneous electron transfer rate because of unsaturated sulfur on the edge sites and stronger quantum confinement. As shown in Figure 1B, the biosensor for glucose detection was fabricated by sequentially growing MoS_2_ nanocrystals and Cu submicron-buds on graphene paper (GP) via hydrothermal and electrodeposition method to form GP-MoS_2_-Cu biosensor. Further coating of lactate oxidase (LOD) on the GP-MoS_2_-Cu electrode, GP-MoS_2_-Cu-LOD biosensor for lactate detection was obtained. Due to the electron transport property and high specific surface area of GP, enhanced catalytic activities, fast electron transfer rate and biosafety of MoS_2_ nanocrystals, the electrochemical biosensor showed excellent sensing performances. For glucose, the electrochemical biosensor had a linear range of 5~1775 μM with a detection limit of 500 nM (S/N = 3). For lactate, the electrochemical biosensor had a linear range of 0.01~18.4 mM with a detection limit of 0.1 μM (S/N = 3).

In order to further enhance the catalytic performances of MoS_2_ nanocrystals, Zhang et al. [71] have incorporated of a secondary metal sulfide (CoS_2_) into MoS_2_ nanocrystals to obtain binary metal sulfide composites (CoS_2_-MoS_2_). Based on CoS_2_-MoS_2_, a non-enzymatic electrochemical biosensor for determination of ascorbic acid, dopamine and nitrite have been proposed with linear ranges of 9.9~6582, 0.99~261.7 and 0.5~5160 μM, respectively. In addition, due to its good electrochemical activity caused by synergistic effect between CoS_2_ and MoS_2_, low detection limits of the electrochemical biosensor for determining ascorbic acid (3.0 μM), dopamine (0.25 μM) and nitrite (0.20 μM) have obtained, respectively.

#### 2.2.2. PEC Biosensors

Due to their superior photoactivities and conductivity, metallic sulfide nanomaterials have been used in various PEC biosensing systems to be photoactive materials or be one of other components through combining with photoactive materials [72,73,74,75,76,77].

Guo and Liu et al. [50] have fabricated a PEC biosensor for detecting a breast cancer biomarker human epidermal growth factor receptor-2 (HER2) (Figure 2A) using WS_2_ nanowire on Ti mesh (WS_2_ NW/TM) as photoactive material. Under visible light excitation, photo energy collected by WS_2_ NW/TM electrode was higher than that of its bandgap. Accordingly, electron was transferred from valence band (VB) to conduction band (CB), and then CB electron was transferred to the surface of Ti mesh, finally the hole in VB was scavenged by H_2_O_2_. Based on the electron transfer process, photocurrent was generated. Moreover, to obtain a dual signal PEC amplification strategy, AuNPs modified with glucose oxidase (GOx) and HER2 specific peptide for signal amplification were utilized. The localized surface plasmon resonance (LSPR) of AuNPs generated a collective oscillation of free electrons when excited by visible light. The free electrons can transfer from Au to the CB of the WS_2_ NW/TM electrode, which enhanced the photoelectric transfer efficiency and then achieved dual signal amplification. GOx modified in AuNPs catalyzed glucose to produce H_2_O_2_, which scavenged the hole in VB of the WS_2_ NW/TM electrode.

For binding with HER2 molecules, HER2 aptamers were modified on the WS_2_ NW surface via oxygen containing sulfur species of WS_2_ NW. The HER2 specific peptides modified on the surface of AuNPs were also utilized to bind with HER2 molecules. When detected HER2 molecules, a sandwich type dual signal PEC amplification biosensor was established with a wide linear range (0.5~10 ng/mL) and low detection limit (0.36 ng/mL, S/N = 3).

Cui et al. [78] have also reported a PEC biosensor for determination of polynucleotide kinase (PNK) based on Bi_2_S_3_ nanorods as the photoactive materials (Figure 2B). The Bi_2_S_3_ nanorods displayed photoactive properties and generated a high photocurrent when excited by visible light. For fabricating PNK biosensor, a hybrid film consists of Bi_2_S_3_ nanorod and AuNPs was used to modify ITO electrode and to bind with capture probe (P1). Manganese based mimic enzymes (MnME) were modified with AuNPs to obtain MnME@AuNPs composites, which could label signal probes (P2). The capture probe on the modified electrode can specifically hybridize with the MnME@ AuNPs-labeled signal probe to form a double-stranded DNA. In the absence of PNK, MnME can catalyze H_2_O_2_ with 3,3-diaminobenzidine (DAB) as substrate, and generated MnME catalytic precipitations on the modified ITO electrode. The MnME catalytic precipitations were insulating barriers and blocked the interfacial electron transfer and eventually leaded to a low PEC signal. In the presence of PNK, the double-stranded DNA was phosphorylated and subsequently cleaved by lambda exonuclease to release the MnME@AuNPs from the modified electrode, leading to a high PEC signal. Based on the signal on-off PEC strategy, the PNK biosensor was proposed and exhibited high sensitivity with a detection limit of 1.27 × 10^−5^ U/mL.

In addition to being photoactive materials, metallic sulfide nanomaterials also have been one of other components through combining with photoactive materials in PEC biosensing systems. For example, Zhao et al. [73] have also reported a PEC biosensor for determination of prostate specific antigen (PSA) using CdTe/TiO_2_ sensitized structures as photoactive materials and CuS nanocrystal as electronic extinguisher (Figure 2C). For fabricating the PSA biosensor, a peptide was fixed to the CdTe/TiO_2_ electrode surface and used to immobilize a double-helix DNA (dsDNA). Then, CuS nanocrystal was efficiently immobilized on the dsDNA via doxorubicin (Dox) inserting into the dsDNA. In absence of PSA, electron donor and radiant light were consumed by CuS nanocrystals, and steric hindrance effect of insulating substances (e.g., peptides and DNA) generated, leading to a low PEC signal. In the presence of PSA, the PSA specifically cleaved the peptide, and DNA/Dox-CuS probes were released from the electrode surface, resulting in a high PEC signal. Take advance of the signal on-off PEC strategy, the PSA biosensor revealed good sensing performance with a linear range from 0.005 to 20 ng/mL and a low detection limit of 0.0015 ng/mL.

#### 2.2.3. ECL Biosensors

Due to their ECL properties, metallic sulfide nanomaterials have been used to establish ECL biosensors. For example, Babamiri et al. [70] have prepared an ECL biosensor for determining human immunodeficiency virus (HIV) DNA sequence utilizing EuS nanocrystals as ECL luminophore through a molecularly imprinted polymer ECL (MIP-ECL) system (Figure 3A). In the MIP-ECL system, HIV aptamer as template and o-phenylenediamine as the functional monomer were electropolymerized directly on the surfaces of the ITO electrode. After removing HIV aptamer template, the MIP modified electrode was obtained. The MIP modified electrode can bind with HIV-1 gene when immersed into different concentrations of HIV-1 gene standard solution. Then, the HIV-1 gene on the MIP modified electrode reacted with the HIV DNA strand functionalized on EuS nanocrystals by hybridization reaction. Based on the hybridization reaction between HIV-1 gene and HIV DNA strand, the MIP-ECL biosensor was proposed. Using K_2_S_2_O_8_ as co-reactant, the ECL signal of the MIP-ECL biosensor significantly enhanced with increased concentrations of HIV-1 gene. Taking advantage of both MIP-ECL assays and the ECL properties of EuS nanocrystals, the HIV gene biosensor was sensitive and selective with a wide linear range (3.0 fM~0.3 nM) and low detection limit (3.0 fM).

Moreover, Zhu et al. [79] have also fabricated a sandwich-type ECL biosensor for detecting insulin based on the ECL property of zinc-doping cadmium sulfide (Au-ZnCd_14_S) (Figure 3B). Au-ZnCd_14_S combined nitrogen doping mesoporous carbons (Au-ZnCd_14_S/NH_2_-NMCs) acted as sensing platform and Au-Cu alloy nanocrystals were employed as labels to quench the ECL of Au-ZnCd_14_S/NH_2_NMCs. On the basis of the ECL quenching effects between ZnCd_14_S and Au-Cu alloy nanocrystals, a sensitive ECL immunosensor for insulin detection was successfully constructed with a linear response range from 0.1 pg/mL to 30 ng/mL and detection limit of 0.03 pg/mL (S/N = 3). Although some metallic sulfide nanomaterials did not exhibit ECL properties, they have also been used to construct ECL biosensors via being as electrode materials based on their superior conductivity and large specific surface area. For example, Wei et al. [80] have reported an ECL biosensor for detection of amlodipine besylate (AML) based on reduced graphene oxide-copper sulfide (rGO-CuS) composite coupled with capillary electrophoresis (CE) (Figure 3C). The rGO-CuS composite was synthesized based on flowerlike CuS wrapped with rGO sheet and utilized to modify electrode. Due to the presence of rGO-CuS composite, the electron transfer rate between the electroactive center of Ru (bpy)_3_^2+^ and the electrode was facilitated. At the present of AML, the ECL intensity of Ru (bpy)_3_^2+^ increased which induced the development of AML biosensor. Take advance of large specific surface area of rGO-CuS composite and powerful CE separation technique, the ECL biosensor for the detection of AML was successfully fabricated with a linear response range of 0.008 to 5.0 μg/mL and a detection limit of 2.8 ng/mL (S/N = 3).

Moreover, Xue et al. [52] have also designed a procalcitonin (PCT) biosensor based on dual-quenching ECL-RET strategy utilizing hollow Ru-In_2_S_3_ nanocomposite as ECL acceptor and porous α-MoO_3_-Au structure as ECL donor (Figure 3D). Specifically, Ru-In_2_S_3_ nanocomposite was prepared by hollow In_2_S_3_ nanotubes as substrate adsorbing Ru (bpy)_3_^2+^. For fabricating PCT biosensor, HWRGWVC heptapeptide (H7), which could provide -SH, was immobilized on the surface of nanomaterials through amide bond (with Ru-In_2_S_3_ nanocomposite) and Au-S bond (with α-MoO_3_-Au structures) and used to capture antibody (Ab_1_ and Ab_2_). In the presence of PCT, Ru-In_2_S_3_ nanocomposite captured Ab_1_ and α-MoO_3_-Au structures captured Ab_2_ connected together, and ECL-RET from Ru-In_2_S_3_ to α-MoO_3_-Au occurred which was further confirmed by testing the overlap between ECL emission of Ru-In_2_S_3_ and UV-vis spectra of α-MoO_3_-Au. Take advantage of huge specific surface area of Ru-In_2_S_3_ or α-MoO_3_-Au and dual-quenching ECL-RET strategy, the ECL biosensor for detecting PCT was obtained with sensitive response, linear range from 0.0001 to 50 ng/mL and low detection limit of 12.49 fg/mL (S/N = 3).

#### 2.2.4. PL Biosensors

Metal sulfide nanomaterials also have been used to establish PL biosensors due to their fascinating optical properties. However, to our best well know, metal sulfide nanomaterials used to fabricate PL biosensors mainly were 0-D metal sulfide nanomaterials (namely, sulfide quantum dots). Thus, PL biosensors based on metal sulfide nanomaterials will be illustrated in the section of “sulfur-containing quantum dots” below.

As described above, biosensors based on metal sulfide nanomaterials have been used for detection of various analytes, including glucose, dopamine, proteins, DNA, etc. Moreover, these biosensors displayed good sensing performance toward analytes detection. In addition, these biosensors also showed other outstanding advantages, including simple of preparation, low cost and good selectivity, stability, and great promising practical applications in clinical diagnosis, as shown in Table 2.

## 3. Sulfur-Containing Quantum Dots

### 3.1. Generalities

Sulfur-containing quantum dots are quantum dots containing central sulfur-containing nanodots and surface functional groups (e.g., carboxyl groups or amino groups), and possess fascinating photophysical properties, small size (typically below 10 nm), good biocompatibility, and chemical inertness. They can be broken down into three main categories: sulfur quantum dots, sulfide quantum dots and sulfur-doped quantum dots.

Sulfur quantum dots. Sulfur quantum dots are pure elemental quantum dots, mainly including S central nanodots and surface functional groups [81,82].

As a new class of quantum dots, sulfur quantum dots were firstly synthesized by Li’s group through phase interfacial reactions in 2014 [83]. Since then, researchers have eagerly pursued synthetic approaches of sulfur quantum dots due to their excellent aqueous dispersibility, small size, excellent photostability, low toxicity, narrow size distribution and ultrahigh photostability [84]. To date, sulfur quantum dots have not been available in the market. Generally, sulfur quantum dots were synthesized by hydrothermal methods based on “top-down” synthetic approaches. Literature has reported detailed synthetic approaches including phase interfacial reaction [81,83], “assemble-fission” approach [82,85], H_2_O_2_-assisted “top-down” approach [86] and oxygen accelerated scalable approach [84]. Synthetic details for each approach are described as follows:

For phase interfacial reaction, CdS quantum dots or ZnS quantum dots were diluted by n-hexane and then sonicated to form a homogeneous solution. HNO_3_ aqueous solution was mixed with CdS quantum dots or ZnS quantum dots solution with a slowly stirring at room temperature. The resulting white mixture was separated by a funnel, and sulfur quantum dots were synthesized as a white suspension in hexane.

For “assemble-fission” approach, sulfur quantum dots were synthesized by simply treating sublimated sulfur powders with alkali using polyethylene glycol-400 as passivation agents.

For H_2_O_2_-assisted “top-down” approach, sulfur quantum dots were synthesized by dissolved bulk sulfur powder into small particles in an alkaline environment in the presence of polyethylene glycol, followed by the H_2_O_2_-assisted etching of polysulfide species.

For oxygen accelerated scalable approach, sulfur quantum dots were synthesized by dissolved bulk sulfur powder into small particles in an alkaline environment in the presence of polyethylene glycol to form polysulfide species (S_x_^2−^), followed by oxidation of S_x_^2−^ to zero-valent sulfur under a pure O_2_ atmosphere sulfide quantum dots. Sulfide quantum dots commonly included central sulfide nanodots, especially metal sulfide, and surface functional groups. Much research efforts have been devoted to synthesize sulfide quantum dots, such as ZnS, CdS, PbS, Ag_2_S, SnS_2_, In_2_S_3_ and AgInZnS quantum dots [87,88,89,90,91,92].

Sulfide quantum dots were commonly synthesized by simple aqueous method and used various stabilizing agents or sulfides to maintain metal atoms in order to assembly into nanodots. The stabilizing agents or sulfides were surface ligands and S sources, and the stabilizing agents included cysteamine, mercaptoacetic acid, l-cysteine, N-acetyl-l-cysteine, bovine serum albumin [67,69,87,93,94,95]. To date, various of sulfide quantum dots, such as PbS quantum dots, Ag_2_S quantum dots and CdSeS/ZnS quantum dots, have been available in the market.

Sulfur-doped quantum dots. Sulfur-doped quantum dots are obtained by doping S atoms into other quantum dots, such as silicon, carbon, phosphorus and graphene quantum dots [82]. Among sulfur-doped quantum dots, sulfur-doped carbon or graphene quantum dots were the most widely studied in the recent years [96,97,98,99,100,101]. This review will focus on sulfur-doped carbon or graphene quantum dots.

The approaches used to synthesize sulfur-doped carbon or graphene quantum dots can be divided into two categories: “top-down” approaches and “bottom-up” approaches. The “top-down” approaches included hydrothermal, solvothermal, ultrasound, chemical exfoliation, microwave-assisted exfoliation methods, and so on [102,103,104,105,106]. Due to their superiority such as time-saving and easy to operation, the “top-down” approaches have attracted much excitement for synthesizing sulfur-doped carbon or graphene quantum dots. The “bottom-up” approaches used to synthesize sulfur-doped carbon or graphene quantum dots can be controlled by “step-by-step” chemical reactions through various precursors [107,108]. To date, carbon or graphene quantum dots have been available in the market, but sulfur-doped carbon or graphene quantum dots haven’t been available in the market yet.

### 3.2. Applications in Biosensors

Sulfur-containing quantum dots are considered to be suitable alternative nanomaterials in biosensing applications [67,109,110,111,112]. Their stable photoelectric properties made sulfur-containing quantum dots be adapted as excellent probes in biosensors via various strategies, such as electrochemical, PEC, PL and ECL strategy [113,114,115]. Soluble sulfur-containing quantum dots can react with biomolecules, thus biosensors for detection biomolecules could be established through specific physiochemical reactions between them [81,116,117]. Functionalization of sulfur-containing quantum dots (especially, sulfide quantum dots) with different stabilizing agents to form surface groups can enhance their hydrophilicity and interaction ability with other biomolecules [115,118,119,120]. The low toxicity of sulfur-containing quantum dots made them suitable to be used for sensing in cells or living bodies [83,121,122,123].

#### 3.2.1. Biosensors Based on Sulfur Quantum Dots

As emerging quantum dots, sulfur quantum dots have been paid much attention due to their possessions of inexpensive S atoms and unique physicochemical properties [82,83,84]. Literature has demonstrated that sulfur quantum dots were applied in the field of sensor [81,85,124]. For example, sulfur quantum dots have been used for sensing metal ions or detecting drug [81,125]. Very recently, sulfur quantum dots also have gradually been applied to living cells imaging [124]. However, applications of sulfur quantum dots in biosensing or bio-medical diagnosis field were still far from satisfactory. To this end, there is an urgent need of efficient approaches to exploit biosensing applications of sulfur quantum dots in the next few years.

#### 3.2.2. Biosensors Based on Sulfide Quantum Dots

Due to optical responses of sulfide quantum dots from the visible to the near infrared (NIR), sulfide quantum dots have received extensive attention in the field of biosensing [8,126]. They have been widely used as alternative probes for biomolecules via various strategies, such as electrochemical, PEC, ECL and PL strategies [94,109,126,127].

##### Electrochemical Biosensors

Due to excellent electrochemical activities of sulfide quantum dots and inexpensive instruments and simple operations of electrochemical methods, electrochemical biosensors based on sulfide quantum dots have attracted increased attention. Zhang et al. [128] have reported an electrochemical biosensor for detecting clenbuterol antibody based on ZnS quantum dots. Amor-Gutiérrez et al. [129] have established an electrochemical biosensor for determination bacteria based on Ag_2_S quantum dots.

##### PEC Biosensors

PEC sensing, a branch of electrochemistry, is a newly developed technology and has attracted great interest in biosensing fields. For fabricating PEC biosensor, photoactive materials are vital because they can generate photocurrent excited by light. To our best well know, sulfide quantum dots not only have directly been photoactive materials to establish PEC sensors [89,92,130,131,132], but also been one of other components through combining with photoactive materials to indirectly establish PEC sensors [87,133].

Wang et al. [130] have proposed a PEC biosensor for detection of H_2_S released from MCF-7 cells based on heterostructures formed by CdS quantum dots and branched TiO_2_ nanorods (CdS-B-TiO_2_). Herein, CdS-B-TiO_2_ heterostructures in the PEC biosensors were directly as photoactive materials. In addition, due to the formation of CdS-B-TiO2 heterostructures, a significant enhancement in photocurrent was obtain, thus leading to sensitive PEC recording of the H_2_S level in cellular environments.

Moreover, Deng et al. [133] have utilized CdS quantum dots as one of other components through combining with photoactive materials to indirectly establish PEC biosensors for determination of PSA. The PSA biosensor was utilized reduced graphene oxide-TiO_2_ (ERGO-TiO_2_) as reduced graphene oxide-TiO_2_ (ERGO-TiO_2_) and CdS quantum dots as a PEC signal amplifier. For preparing the PSA biosensor, ERGO-TiO_2_ was utilized to immobilize capture antibody (Ab1) for PSA detection, and quinone-rich PDA nanospheres (PDANS) loaded with CdS quantum dots were used to load detection antibody (Ab2) for PSA detection. In the presence of PSA, photo-generated electron transferred between PDANS loaded with CdS quantum dots and ERGO-TiO_2_. Due to the good conductivity of PDANS, ERGO and CdS quantum dots, a PSA biosensor has been proposed with a linear range from 0.02 pg/mL to 200 ng/mL with the detection limit of 6.8 fg/mL.

##### ECL Biosensors

Sulfide quantum dots have been widely used as alternative probes for biomolecules (e.g., dopamine, thrombin, laminin or enzyme) via ECL strategies.

Liu et al. [134] have fabricated a dopamine (DA) biosensor based on water-dispersible CdS quantum dots (CdS QDs) via ECL strategy. As shown in Figure 4A, they synthesized four sizes of CdS QDs, namely 1.8, 2.7, 3.2 and 3.7 nm. Each size of CdS QDs had various ECL performance. Under the optimized conditions, the ECL biosensor displayed excellent sensing properties with linear detection range from 8 pM to 20 nM and detection limit of 3.6 pM (S/N = 3).

In addition, Wang et al. [135] have also proposed a thrombin (TB) biosensor based on lanthanum ion-doped CdS quantum dots (CdS: La QDs) via ECL strategy (Figure 4B). The detection mechanism of the ECL biosensor was based on a distance-dependent ECL intensity enhanced or quenched system between CdS: La QDs and AuNPs. In the presence of Hg^2+^, ECL quenching (signal off) achieved lie in RET between the CdS: La QDs and AuNPs at a close distance. In the presence of TB, ECL enhance (signal on) achieved lie in surface plasmon resonances (SPR) between the CdS: La QDs and AuNPs at a separated distance. The “on-off-on” approach was used to detect TB, and the linear range were 1.00 × 10^−16^ to 1.00 × 10^−6^ mol/L with limit of detection (S/N = 3) of 3.00 × 10^−17^ mol/L.

Moreover, Wu et al. [136] have prepared a laminin (LN) biosensor based on Mn doped Ag_2_S quantum dots (Ag_2_S: Mn QDs) as ECL materials (Figure 4C). The optical response of Ag_2_S: Mn QDs was in IR window (i.e., about at 626 nm) obtained by ECL spectrum. Based on a sandwiched ECL immunoassay, the biosensor displayed a wide linear range of 10 pg/mL to 100 ng/mL with a low detection limit of 3.2 pg/mL for LN detection.

Furthermore, Zhou et al. [137] have proposed an enzyme (namely, DNA methyltransferase (MTase)) biosensor based on CdS quantum dots (CdS QDs) as ECL materials (Figure 4D). For fabricating the MTase biosensor, double-stranded DNA containing 5′-CCGG-3′ sequence was bonded to a CdS QDs modified glassy carbon electrode, then the modified electrode was incubated with M.SssI CpG MTase which catalyzed the methylation of the specific CpG dinucleotides. Subsenquently, the electrode was treated with a restriction endonuclease HpaII. The HpaII can recognize and cut off the 5′-CCGG-3′ sequence, but recoginition function was blocked when the CpG site in the 5′-CCGG3′ was methylated. Double-stranded DNA having been methylated can immobilized AuNPs with glucose oxidase mimicking activity. AuNPs immobilized on double-stranded DNA can catalyze the oxidation of glucose to genetate H_2_O_2_ which served as coreactant of CdS QDs. Thus, the ECL intensity of CdS QDs was linear correlation with the activity of M.SssI MTase.

##### PL Biosensors

Sulfide quantum dots, as 0-D metal sulfide nanomaterials, have also attracted great attention to establish PL biosensors due to their fascinating optical properties [138]. Until now, there have been various types of PL biosensors based on sulfide quantum dots, such as phosphorescence biosensors, fluorescence biosensors.

Phosphorescence biosensors were proposed based on sulfide quantum dots with phosphorescence emission. For example, Gong and Fan [139] have proposed a phosphorescence biosensor for detection of DNA based on riboflavin-modulated Mn doped ZnS quantum dots. Due to the longer average life of phosphorescence emitted by Mn doped ZnS quantum dots, the DNA biosensor allowed appropriate delay time and avoided any scattering light.

Because most of sulfide quantum dots have fluorescent properties, fluorescence biosensors based on sulfide quantum dots were the most common biosensors. For example, Liu et al. [140] have prepared a fluorescence biosensor for detecting alkaline phosphatase based on l-cysteine-capped CdS QDs. Moreover, Du et al. [67] have synthesized VS_2_ quantum dots, and constructed a glutathione biosensor based on the efficient fluorescence RET from VS_2_ quantum dots to MnO_2_ nanosheets and fast redox reaction between MnO_2_ and glutathione. Adegoke et al. [141] have utilized CdZnSeS/ZnSeS quantum dots to fabricate a fluorescence biosensor for determining influenza virus RNA. Rong et al. [142] have synthesized novel Eu^3+^ ion-functionalized fluorescent MoS_2_ quantum dots for biosensing Guanosine 3′-diphosphate-5′-diphophate (ppGpp) (Figure 5A). Literature has also reported fluorescence biosensors for determining other biomolecules, such as thrombin, glutathione S-transferase enzyme, bilirubin [91,94,143,144] via PL strategies.

NIR fluorescence biosensors were emerging types of fluorescence biosensors and have also attracted great attention. The NIR fluorescence biosensors were fabricated based on fluorescence materials with emissions in NIR window (750 to 1000 nm) which can achieve higher imaging depth without complications from tissue autofluorescence. Sulfide quantum dots, such as Ag_2_S quantum dots, displayed fluorescence emission in NIR window, therefore they have been used to establish biosensors. Moreover, Ding et al. [123] have fabricated a NIR biosensor for detecting F^−^ in living cells based on NIR emitting Ag_2_S quantum dots. The fluorescence intensity of Ag_2_S quantum dots enhanced when various rare earth ions were added. In the presence of F^−^, F^−^ coordinated with rare earth ions leaded to fluorescence quenching of Ag_2_S quantum dots. Based on the on-off fluorescence findings, a label-free NIR fluorescence biosensor for F^−^ in living has been proposed. Moreover, Shu et al. [69] have ameliorated Ag_2_S quantum dots by doping Pb ions to synthesized Pb-doped Ag_2_S quantum dots. The Pb-doped Ag_2_S quantum dots emitted fluorescence in NIR-Ⅱ window (950 to 1200 nm). Based on the NIR-Ⅱ emitting Pb-doped Ag_2_S quantum dots, a biosensor for H_2_O_2_ have been proposed (Figure 5B).

#### 3.2.3. Biosensors Based on Sulfur-Doped Carbon or Graphene Quantum Dots

Carbon or graphene quantum dots have attracted intensive interest and have also been used to determine biomolecules due to their fascinating properties [145,146,147,148]. Doping heteroatoms (e.g., nitrogen, sulfur, phosphorus/or metal atoms) in carbon or graphene quantum dots is an effective way to tune their properties [107,112,149,150]. As the third most abundant element in fossil fuels, S and its derived material have attracted a lot of interest. Sulfur doped carbon or graphene quantum dots, as one of derived sulfur-containing nanomaterials, have also attracted intense interest and been widely used to established biosensors [151,152,153,154]. However, sulfur doped carbon or graphene quantum dots still belonged to carbon or graphene quantum dots. Since too much literature has reported biosensors based on carbon or graphene quantum dots [155,156,157,158], we won’t explore them in this review. As described above, biosensors based on sulfur-containing quantum dots have been used for detection of various analytes, including antibody, dopamine, proteins, DNA, RNA, glutathione, bacteria, F^−^ in living cells, etc. These biosensors displayed good sensing performance toward analytes detection. In addition, these biosensors also showed other outstanding advantages, including simple of preparation, low cost and good selectivity, stability, and great promising practical applications in clinical diagnosis, as shown in Table 3.

## 4. Brief Comparison between Biosensors Based on Sulfur-Containing Nanomaterials and Others

All in all, biosensors based on sulfur-containing nanomaterials have been used for detection of various biomolecules in the last five years, including glucose, dopamine, proteins, DNA, RNA, etc. The biosensors based on sulfur-containing nanomaterials displayed enhanced selectivity, lower sensitivity, faster response time, and low detection limit in comparison to biosensors based on other nanomaterials. Taking glucose as analytes, Table 4 displays brief comparison between biosensors based on sulfur-containing nanomaterials and other biosensors. Data listed in Table 4 illustrates that sulfur-containing nanomaterials are promising materials to established biosensors and can be widely used in biomedical field.

## 5. Conclusions and Outlooks

In summary, this paper provides a brief overview of recent researches on the applications of sulfur-containing nanomaterials, including metallic sulfide nanomaterials and sulfur-containing quantum dots in biosensors. The sulfur-containing nanomaterials have excellent properties, such as nanometric scale, water-dispersibility, excellent catalytic activity, conductivity, biosafety, photoactivity, and fascinating optical properties, and have been proven useful in various biosensing applications via electrochemical, PEC, ECL and PL strategies. Though many achievements have been obtained for biosensors based on sulfur-containing nanomaterials, there are still significant challenges that need to be solved.

(1) As an emerging quantum dots, sulfur quantum dots possess excellent optical properties and biocompatibility which make them possible to prepare biosensors. However, researches on biosensing applications of sulfur quantum dots are still inadequate. Therefore, it is urgent to further exploit the biosensing applications of sulfur quantum dots in the next few days. Taking advantage of optical properties of sulfur quantum dots and PL-based technologies (e.g., fluorescence detection technologies), PL probes for detecting various biomolecules based on sulfur quantum dots can be established. (2) Real-time biosensing in vivo or intracellular based on sulfur-containing nanomaterials remains a challenge because typical analytical measurements only capture a single-time-point in samples. Biosensing in vivo or intracellular are a new class of detecting technologies that can be established by means of a number of sophisticated analysis platforms providing an in vivo read-out of the spatial, temporal, and quantitative information of biomolecules. Therefore, the vast majority of analytes detected by biosensors based on sulfur-containing nanomaterials are limited to exist in vitro or extracellular. In order to fabricate vivo or intracellular biosensors based on sulfur-containing nanomaterials, sophisticated analysis platforms providing real-time information of biomolecules should be tried to fabricate by utilizing various of technologies and methods (such as, confocal fluorescence microscopic techniques and Raman spectroscopy methods).

(3) With the development of materials science and nanotechnology, wide variety of nanomaterials have emerged in our life. Some nanomaterials, such as metal organic frameworks and gold nanoclusters, are easy to synthesize without complicated operations. Considering time cost and experimental safety, more and more researchers have dedicated to exploiting these nanomaterials. In view of the abundant storage and the pressure on the environment of S elements, more and more sulfur-containing nanomaterials should be synthesized and used to construct biosensors. Therefore, green synthetic methods (e.g., coprecipitation and hydrothermal methods) should be exploited and utilized to synthesize sulfur-containing nanomaterials.

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
