# Peer review of "Biosensors Based on Advanced Sulfur-Containing Nanomaterials"

_sensors, 2020, doi:10.3390/s20123488_

Round 1

Reviewer 1 Report

The manuscript represents comprehensive review on application of sulfur-containing nanomaterials in development of biosensors. The review is well written and contains newest achievements in this field including own works of authors. It is in scope of Sensors and can be published after minor improvements:

  1. Briefly mention methods of preparation sulfur-containing materials, especially those based on quantum dots. Explain whether some of them are available in the market.
  2. Would be useful to provide brief comparative analysis on advantage of biosensors based on sulfur-containing materials with those of other approaches, for example by selecting one-two most common analytes and preparing table in which basic properties such as limit of detection, linear range of detection can be shown and compared.
  3. Please mention what is effectivity of application of biosensors based on sulfur-containing materials in detection of analytes in real sample. Discuss the sensors recovery

Author Response

Dear reviewer,

We highly appreciate the detailed valuable comments of the reviewer on our manuscript of “Biosensors based on advanced sulfur-containing nanomaterials” (sensors-831055). These suggestions are quite helpful for us and we incorporate them in the revised manuscript. We have made a point-by-point response to the comments raised by reviewer. Comments of reviewer are written in black, responses of the authors in red and instructions for changes in blue. Meanwhile, we have provided the revised manuscript that have all changes clearly highlighted using the “Track Changes” function in response to these comments clearly.

If further information is needed, please kindly let us know it soon at your earliest convenience. Thank you very much in advance for all your kind assistance.

Sincerely Yours

Xuemei Wang

Reviewer 2 Report

See attached file 

Author Response

(The authors gave the same response as above.)

Reviewer 3 Report

This review paper by Li et al. reports on research and development of sulfur-containing nanomaterials for biosensor applications. The authors nicely summarized the recent progress made in the field of some selected class of sulfur-related nanomaterials along with future perspective and challenges for wide use of the material in biosensor applications.

              The manuscript is well-organized and can be a good contribution to the field of nanomaterials. I will recommend its publication after the authors consider the following points:

  1. The authors selected “two categories of sulfur-containing nanomaterials”. It is better explained why they chose the class of material. It will be also helpful for readers if they can give brief information about other materials together with recent review articles.
  2. Conclusions and outlooks section can be improved by adding possible solution to those challenges stated. For example, what strategy can be possible to put sulfur quantum dots in practical use as biosensors; what makes in vivo or intracellular biosensor application difficult and how the challenges can be overcome; why sulfur-containing nanomaterials are not synthesized so much and how to improve the situation etc. These points should also include comparison to other materials and devices already used in practical scenes.
  3. English writing needs to be improved. For instance, “an emerging types” on page 13 should be “emerging types”. On the same page, “… in living have been …” should be “… in living has been …”. There are similar grammatical errors throughout the manuscript.

Author Response

(The authors gave the same response as above.)
